# Imaging Techniques for Aortic Aneurysms and Dissections in Mice: Comparisons of Ex Vivo, In Situ, and Ultrasound Approaches

**DOI:** 10.3390/biom12020339

**Published:** 2022-02-21

**Authors:** Sohei Ito, Hong S. Lu, Alan Daugherty, Hisashi Sawada

**Affiliations:** 1Saha Cardiovascular Research Center, College of Medicine, University of Kentucky, Lexington, KY 40536, USA; sohei.ito@uky.edu (S.I.); hong.lu@uky.edu (H.S.L.); alan.daugherty@uky.edu (A.D.); 2Saha Aortic Center, College of Medicine, University of Kentucky, Lexington, KY 40536, USA; 3Department of Physiology, College of Medicine, University of Kentucky, Lexington, KY 40536, USA

**Keywords:** imaging approach, aortic diseases, aortopathy, mouse

## Abstract

Aortic aneurysms and dissections are life-threatening conditions that have a high risk for lethal bleeding and organ malperfusion. Many studies have investigated the molecular basis of these diseases using mouse models. In mice, ex vivo, in situ, and ultrasound imaging are major approaches to evaluate aortic diameters, a common parameter to determine the severity of aortic aneurysms. However, accurate evaluations of aortic dimensions by these imaging approaches could be challenging due to pathological features of aortic aneurysms. Currently, there is no standardized mode to assess aortic dissections in mice. It is important to understand the characteristics of each approach for reliable evaluation of aortic dilatations. In this review, we summarize imaging techniques used for aortic visualization in recent mouse studies and discuss their pros and cons. We also provide suggestions to facilitate the visualization of mouse aortas.

## 1. Introduction

Aortic aneurysms are defined as a permanent dilatation of the aortic wall that occurs commonly in the infrarenal abdominal or proximal thoracic region [1,2,3]. Aortic dissections are a tearing of the aortic wall, predominantly initiated in the thoracic region [1,2,3,4]. Aortic rupture and organ malperfusion are lethal consequences of these diseases [2,3,4]. There is no validated pharmacological approach to prevent or reduce aortic aneurysms and dissections (AADs). Thus, animal studies are needed to uncover the molecular basis of AADs for the development of new therapeutics. Over the past two decades, multiple mouse models have been established to investigate the mechanism of AADs. These include angiotensin II infusion, periaortic application of calcium chloride, intraluminal or periaortic elastase infusion, and β-aminopropionitrile administration [5,6,7,8,9,10,11]. Many molecular mechanisms have been investigated using these mouse models.

In humans, AADs are monitored by ultrasonography, computed tomographic (CT) angiography, or magnetic resonance imaging (MRI) [12,13]. In mice, aortic dimensions are evaluated mainly by three approaches; ex vivo, in situ images, and ultrasonography. Ex vivo and in situ approaches can visualize the aorta in any region directly and readily after termination without expensive imaging modality. Therefore, these approaches have been used widely in mouse AAD studies. With the advent of the high-frequency ultrasound system, aortic imaging in mice has evolved rapidly. Ultrasonography enables monitoring of the aorta in living mice [14,15]. Sequential evaluation is a considerable advantage of this mode. These three approaches have provided important insights into understanding the pathophysiology of AADs, while reliable evaluation of AADs in mice is often challenging because of multiple confounding factors, such as morphological alteration of the aorta and the interference by other organs. For authentic assessment of AADs, it is important to understand the strengths and weaknesses of each imaging approach.

In this review, we summarize the primary imaging modality for aortic visualization in recent mouse studies for AADs. We highlight ex vivo, in situ, and ultrasound approaches, discuss their pros and cons, and provide suggestions for aortic imaging in mice.

## 2. Preferred Approaches for Aortic Imaging in Mice

To determine imaging approaches that have been used for the determination of AADs in mice, we summarized recent articles using mouse AAD models. Articles were searched in PubMed with three keywords: “aortic aneurysm”, “aortic dissection”, or “aortopathy”, published in *Arteriosclerosis, Thrombosis, and Vascular Biology* (*ATVB*) from 2015 to 2020. There were 131 articles in *ATVB* during the interval, and 32 articles were excluded because those were not research articles (Figure 1A). There were 99 research articles associated with AADs and 78 articles that used AAD mouse models. A total of 49 (63%) and 11 (14%) articles studied abdominal aortic aneurysm (AAA) and thoracic aortic aneurysm (TAA) mouse models, respectively (Figure 1B). Four articles (5%) used aortic dissection (AD) mouse models. Nine articles (12%) investigated two aortic diseases, TAA and AAA (7 articles, 9%) or TAA and AD (2 articles, 3%).

### 2.1. Imaging Approaches for Aortic Aneurysms

In 20 articles investigating TAAs, 13 studies (65%) performed ultrasonography as the primary approach for aortic measurements (Figure 1C) [15,16,17,18,19,20,21,22,23,24,25,26,27]. Direct measurements using ex vivo or in situ images were performed in four studies (20%) [28,29,30,31]. Three studies (15%) evaluated aortic diameters histologically [32,33,34]. Conversely, direct imaging using ex vivo or in situ approach was the most common mode for AAA determination (38 articles, 68%) [28,29,30,31,35,36,37,38,39,40,41,42,43,44,45,46,47,48,49,50,51,52,53,54,55,56,57,58,59,60,61,62,63,64,65,66,67,68], and ultrasonography was the second most common approach (16 articles, 28%, Figure 1D) [25,26,27,69,70,71,72,73,74,75,76,77,78,79,80,81]. In these articles, 63 articles (96%) measured aortic diameters as a parameter to describe the severity of aortic aneurysms.

The difference of preferred modes for aortic imaging between TAAs and AAAs may be based on differences in histological features in these diseases. Adventitial thickening with collagen deposition is a profound pathology in several AAA mouse models, including angiotensin II infusion [82]. Calcification and thrombus formation are also reported in AAA tissues [82,83]. Therefore, AAA structures are relatively maintained in aneurysmal mice even after termination, and it can be evaluated by direct approaches, including ex vivo and in situ approaches. In contrast, several TAA mouse models exhibit thinning of the aortic wall occasionally at the advanced stage of diseases [84,85]. Thus, aortic patency in TAA is often not maintained in mice with the absence of blood pressure after termination. In TAAs, aortic measurements by direct approaches have a potential of underestimation if the aortic patency is not maintained.

Several articles used histological images for aortic measurements. In these articles, aortic diameters were determined by measuring the longest distance between two points on the inner elastic laminae. Aortic perimeters were also measured by tracing the inner elastic laminae. However, measurement of aortic dimensions using tissue sections is unlikely to provide authentic diameters because the sample preparation will affect aortic measurements.

CT and MRI can also evaluate AADs in mice [86,87], but these modalities were used in only three articles of recent *ATVB* publications [18,28,44]. State-of-the-art micro-CT scanners have 0.5 to 5 µm resolutions [88] and enable volumetric measurements with detailed spatial characterization. Thus, we anticipate that micro-CT scanning will be applied more frequently in the future in mouse AAD studies. However, there are several impediments to applying CT imaging as a standard approach. The requirement of a contrast agent is a considerable impediment. In addition, radiation exposure needs to be considered. Although the radiation dose may vary by modalities, scan areas, and resolutions, mice are exposed to radiation at about 0.3 to 0.5 Gy per scan [89]. The lethal radiation dose is 5.0 to 7.6 Gy in mice [89]. Given the lethality and organ damages by radiation, there are impediments to performing serial longitudinal CT scanning. Conversely, MRI does not need radiation. In addition, time-of-flight angiography and black-blood spin-echo sequences can evaluate aortic morphology without contrast enhancement [90,91]. Despite these technological advancements, MRI is not common for routine use because of the expense of the system and duration of image acquisition.

### 2.2. Imaging of Aortic Dissections

Although there were only six articles present in this literature survey, all articles investigating ADs used direct approaches as the primary mode (Figure 1E) [15,21,92,93,94,95]. In these articles, histological analyses were performed to validate the presence of ADs. In contrast to aneurysmal studies, none of the AD studies used aortic diameters to describe the disease severity. The incidence rate is a common mode to describe the development of ADs, but the definition of ADs is variable among studies [96].

There is no standardized mode to describe the severity of ADs. Two recent studies attempted to describe the severity of ADs by measuring either the length or extent of aortic hemorrhage in ex vivo images [95,97]. However, it is not feasible to discern adventitial hemorrhage from mild ADs by ex vivo visualization. Several articles used a scoring scale for AD severity from normal to aortic rupture, while it may be difficult to represent a modest change in this mode. Thus, it is desirable to develop objective and quantitative parameters for the evaluation of ADs. Recently, we reported that a high-frequency ultrasound system can detect false lumen and flap in mice with ADs [15]. Therefore, it is feasible to use ultrasonography for quantitative evaluation of AD formation in mice.

## 3. Imaging Approaches for Aortic Measurements in Mice

Ex vivo and in situ direct visualizations and ultrasonography were common modes for aortic imaging in mice. Of note, in 78 recent *ATVB* articles, most studies used either mode for aortic visualization (Figure 1C–E). Each mode has different strengths and weaknesses. It is crucial to understand the characteristics of each mode for reliable imaging and accurate measurements.

### 3.1. Ex Vivo Imaging

The ex vivo approach can assess aortic morphology readily after termination without the need for expensive imaging modalities (Figure 2A). This approach can measure aortic dimensions at any region from the proximal thoracic to the distal abdominal aorta but is most commonly used for measurements of AAAs. Since ex vivo mode is optimal to evaluate the gross appearance of aortas, the presence of ADs and rupture is usually assessed by this mode. Despite the ease of use, there is a shortcoming that aortic measurements in this mode are affected by the lack of intraluminal pressure of the aorta. After termination, the intraluminal pressure is lost, which may cause underestimation of aortic diameters, especially in thoracic aortas with severe aneurysmal dilatations. Several articles used either formalin or latex perfusion in physiological pressures during terminations to overcome this shortcoming [95,97,98,99,100,101].

Dissection microscopy is used routinely for ex vivo imaging of mouse tissue, but several articles used unique modalities for detail imaging or functional evaluations of the aortic wall. Phased contrast X-ray tomographic microscopy can construct detailed 3-dimensional (3D) aortic images [18,102,103]. Optical coherence tomography with panoramic digital imaging can assess aortic thickness and stiffness in addition to aortic morphology [104].

**Figure 2 biomolecules-12-00339-f002:**
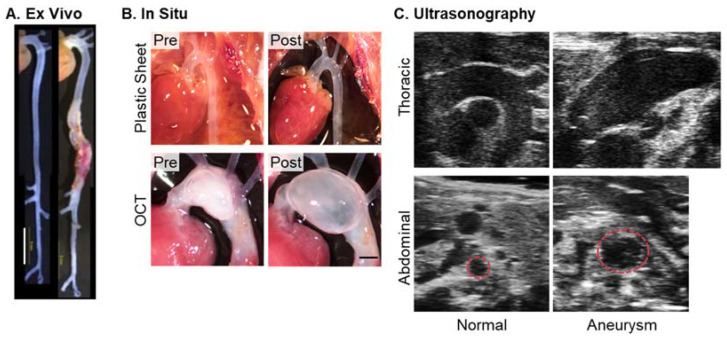
Examples of aortic images in mice. Representative images of (**A**) ex vivo [95] and (**B**) in situ approaches (authors’ unpublished data and the work of [105]), and (**C**) ultrasonography [14]. Ultrasound images were captured by Vevo 2100 with an MS550D (40 MHz). For in situ imaging, a piece of black plastic sheet was placed behind the aorta to enhance the contrast of the aortic wall. Subsequently, O.C.T. (optimal cutting temperature, 100–150 µL) compound was injected from the left ventricle using an insulin syringe (31G). Scale bars in Figure (**A**) and (**B**) indicate 5 and 1 mm, respectively.

### 3.2. In Situ Imaging

In situ approach is less common for aortic visualization compared to ex vivo imaging, but it is a relatively rigorous mode to determine aortic dimensions in mice. In situ method can evaluate the aorta in the anatomical “in situ” position, which may enable more precise evaluation than the ex vivo approach (Figure 2B). However, it is often difficult to visually recognize the aorta in this mode because of the weak contrast of aortic edges (Figure 2B, top left). An approach to address this issue is an insertion of a black plastic sheet behind the aorta, which enhances the contrast of the aortic wall (Figure 2B, top right). Similar to ex vivo imaging, this mode also has the potential for underestimation in aortic measurements due to the absence of blood pressure. Recently, we reported a protocol for maintaining aortic patency by the injection of optimal cutting temperature (O.C.T.) compound [105]. Appropriate injection of O.C.T. does not cause excessive intraluminal pressure or overt tissue damage but maintains aortic patency to provide authentic aortic measurements (Figure 2B, bottom).

Although ex vivo and in situ approaches are practical modes for aortic measurements, these modes evaluate external, not luminal, diameters of the aorta. In AngII-induced AAAs, external and luminal diameters are often different due to adventitial hemorrhages [85,102,106]. In addition, since these approaches are applied after euthanasia, sequential measurements are not feasible in these modes.

### 3.3. Ultrasonography

High-frequency ultrasound is a powerful tool to monitor aortic diameters sequentially in living mice (Figure 2C) [14,107,108]. Therefore, ultrasonography has been used widely in mouse AAD studies. In fact, 33 AAD articles (45%) in recent *ATVB* publications used ultrasonography to measure aortic dimensions. Despite the high versatility, there are several caveats in ultrasonography in terms of its system and settings.

The quality of ultrasound images depends on the ultrasound system, and the ultrasound view affects measurements. Therefore, it is important to describe detailed information of the ultrasound system and settings for rigor and reproducibility. Table 1 demonstrates the number of articles stating this information in recent *ATVB* articles. Most articles (94%) state the ultrasound system used in the articles. The type of ultrasound transducer was stated in 38% of articles, and the specific frequency is described in 62% of articles. Surprisingly, the ultrasound view was stated only in 12% of articles. The information related to the cardiac cycle was described in only 35% of articles.

#### 3.3.1. Ultrasound System

The ultrasound system, including a main unit and transducer, has a considerable impact on image quality. In general, higher frequency provides higher resolution. A high-frequency system is required in the AAD mouse study to detect 0.3–1.0 mm aortic dilatations. [6,85,100,109,110,111] Most studies in recent *ATVB* articles used high-frequency ultrasound systems, such as Vevo 2100 and 3100 (VisualSonics, FUJIFILM) with a high-frequency transducer (MS400: 18–38, MS550D: 22–55, MX550D: 25–55, MS700: 30–70 MHz). In contrast, human ultrasound systems use lower frequency compared to mouse ultrasound systems (2–15 MHz). Thus, it is not recommended to use clinical systems for the visualization of mouse aortas because of insufficient spatial resolution. 

#### 3.3.2. Ultrasound View

Ultrasound view is an important factor for accurate measurements of aortic diameters as aortic diameters may vary by ultrasound views (Figure 3A). The aorta is cylinder-shaped. Thus, the long-axis view has a potential for underestimation, while the short-axis view may cause overestimation. To capture appropriate aortic images, an ultrasound plane must be adjusted to the center and perpendicular to the aorta. For meaningful comparisons, aortic diameters must be measured in the same view and angle. Since ultrasonography is a user-dependent procedure, ultrasound procedures must be standardized for accurate visualizations of mouse aortas.

In the thoracic aorta, the right parasternal long-axis view is optimal for ascending aortic imaging, whereas the left parasternal long-axis view is appropriate for the aortic sinus and proximal ascending aorta [14]. The descending aorta can be visualized by the paraspinal long- and short-axis views [15]. However, ultrasound imaging of the proximal descending aorta is technically difficult due to the interference of the occipital region with the transducer. In the abdominal aorta, ultrasound images have been captured from the ventral side in either a long- or short-axis view [14]. Since AAA formation is not often concentric, in particular the AngII model, the short-axis view is optimal to measure aortic diameters in the abdominal region [107]. The short-axis view can determine aortic areas in addition to dimensions.

#### 3.3.3. Cardiac Cycle

The cardiac cycle affects aortic measurements (Figure 3B). Several articles used the difference of aortic diameters between cardiac phases to calculate aortic stiffness and strain [25,112,113]. To define the cardiac cycle, an electrocardiogram is commonly used during ultrasonography. Most articles measured aortic diameters at either end-diastole or mid-systole. The end-diastole is defined at the R wave, while the mid-systole is technically difficult due to the lack of clarity in the T wave. Therefore, several articles defined the mid-systole when the aorta is maximally and visually dilated. Because of the difference during the cardiac cycle, aortic diameters must be measured at a consistent cardiac phase. Since anesthesia can alter the cardiac function and hemodynamic state, it is also vital to adjust the anesthesia in an appropriate depth.

#### 3.3.4. Three-Dimensional Imaging

Although specific units are required, several ultrasound systems can visualize mouse aortas in 3D [114,115,116]. Three-dimensional imaging enables the evaluation of aortic geometry more precisely, and 3D volumetric measurements can detect aortic dilatations at the earlier time point during AAA formation compared to 2D measurements [108]. Aortic imaging in 3D alleviates many issues associated with probe position and operator dependence. In addition, current ultrasound systems can obtain 3D images over time that assess aortic kinematics and strain [117].

#### 3.3.5. Doppler Ultrasonography

Doppler ultrasound approaches assist the definition of AAD characteristics. Color and power Doppler modes visualize blood flow patterns, and irregular flow patterns have been reported in aneurysmal and dissected mouse aortas [103,118,119]. Pulse wave Doppler can measure flow velocities in aortas, and aortic flow velocities are decreased in several types of aneurysm mouse models [120,121,122]. In addition, pulse wave velocities can be used for the evaluation of aortic stiffness [120,123,124]. Of interest, pulse wave propagation and velocities can be mapped in 2D ultrasound views [125,126].

## 4. Comparisons of Ex Vivo, In Situ, and Ultrasound Approaches

Characteristics of each imaging approach are summarized in Table 2. Direct visualization by ex vivo or in situ approaches does not need expensive devices. In addition, an in situ approach can image aortas in the original anatomical position. However, these modes can only be performed after termination. Ultrasound can measure aortic diameters sequentially in living mice, thereby overcoming the shortcomings of ex vivo and in situ approaches. Ultrasound imaging is becoming more common in mouse AAD studies, but it is technically challenging for the proximal descending aorta. Several mouse models show TAA formation in this region [9,11,97,111]. Thus, it is optimal to perform both in situ and ultrasound imaging for the determination of AADs. The combination of direct approach and ultrasonography compensates for each other and would provide more robust and accurate measurements.

## 5. Limitations

This review has a limitation that articles were searched in a single journal within a restricted period. In this review, *ATVB* was chosen because of its large number of studies on aortic diseases in combination with its emphasis on rigor and reproducibility in preclinical research. To review common approaches for aortic imaging in current mouse studies, articles from 2015 through 2020 in *ATVB* were analyzed. A further review in other journals within an expanded period would enhance the suggestions and recommendations of this review to facilitate reliable evaluations of AADs in mice.

## 6. Summary

This review summarized imaging approaches for the determination of mouse AADs in recent articles published in *ATVB*. Although multiple cutting-edge modalities are available for aortic imaging, ultrasonography, and direct ex vivo or in situ visualization are common techniques in mouse AAD studies. Since uncovering the mechanism of AADs relies on accurate aortic measurements, it is important to understand the feature of each imaging approach for authentic evaluation of AADs.

## Figures and Tables

**Figure 1 biomolecules-12-00339-f001:**
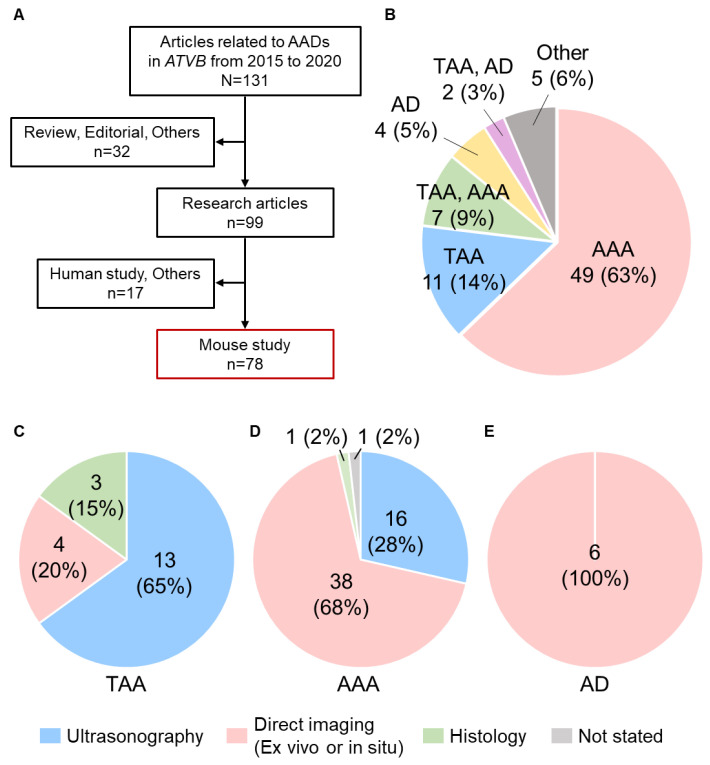
Difference of the primary imaging approach in mouse studies for aortic diseases published in *Arteriosclerosis, Thrombosis, and Vascular Biology* (*ATVB*) from 2015 to 2020. (**A**) Schematic diagram for included or excluded articles in this review. (**B**) The number of articles of aortic studies published in *ATVB* from 2015 to 2020. Primary imaging approaches for the evaluation of (**C**) TAA, (**D**) AAA, and (**E**) AD.

**Figure 3 biomolecules-12-00339-f003:**
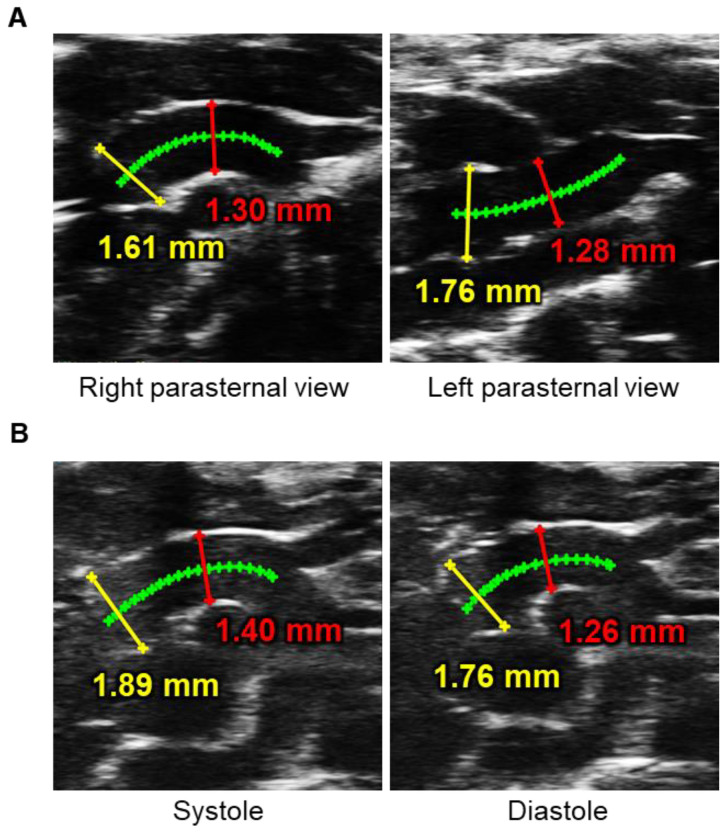
Impacts of ultrasound view and cardiac phase on measurements of aortic diameters in mice. (**A**) Representative ultrasound images of right and left parasternal long-axis approaches in the proximal thoracic aorta. (**B**) Luminal diameters at the aortic root (yellow) and ascending aortic (red) were measured at the mid-systole and end-diastole in the right parasternal long-axis view. Green lines placed on the center of aortic lumen were used as a reference for aortic measurements. Authors’ unpublished data. Ultrasound images were captured by Vevo 3100 with an MX550D (40 MHz).

**Table 1 biomolecules-12-00339-t001:** Number of articles that stated the ultrasound system and settings.

	System	Transducer	Frequency	View	Cardiac Cycle
**Number of Articles (%)**	32 (94)	13 (38)	21 (62)	4 (12)	12 (35)

**Table 2 biomolecules-12-00339-t002:** Comparison of ex vivo, in situ, and ultrasound approaches for aortic imaging in mice.

Approach	Device Cost	Aortic Position	Sequential Imaging	Visible Region
Ex vivo	Low	Artificial	Not feasible	Unlimited
In situ	Low	Physiological	Not feasible	Unlimited
Ultrasound	High	Physiological	Feasible	Limited

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
