# Peer review of "Imaging Techniques for Aortic Aneurysms and Dissections in Mice: Comparisons of Ex Vivo, In Situ, and Ultrasound Approaches"

_biomolecules, 2022, doi:10.3390/biom12020339_

Round 1
Reviewer 1 Report
Ito and colleagues have written a helpful review paper focused on a variety of imaging techniques for the assessment of murine aortic aneurysms and dissections. The authors should be commended for briefly summarizing a variety of in vivo and ex vivo measurement strategies that can be used to assess aortic geometry. However, the choice of both years (2015-2020) and journal (ATVB papers only) limits the impact of this review as numerous older/recent and impactful studies are omitted. While including analysis of all murine AAD studies is likely impossible, expanding the review to include other journals would help the reader understand what is currently possible, at least for the section of imaging strategies (section 3). Further, the works by the authors and others at the University of Kentucky are well-cited (at least 18 citations to previous work from the authors), but key contributions from several other groups are overlooked. None of the references below are required to be added – they are simply included as suggestions. Below are both major and minor comments aimed at helping the authors provide a more complete review.
Major Comments
- Line 50: It is unclear to this reviewer if the AAA group in Figure 1 includes only “true aneurysms” with general lumen expansion or a combination of true and dissecting AAAs (such as those caused by AngII-infusion where the medial and adventitial layers separate in the suprarenal abdominal aorta). If a combination, how does this group differ from the aortic dissection (AD) group also described in and section 2.2? Perhaps a simpler way to separate these AAA and AD studies is by location (i.e. ascending thoracic aorta vs. descending thoracic aorta vs. abdominal aorta). Otherwise, it is important in this reviewer’s opinion to distinguish true AAAs (from elastase and CaCl2 for example) from dissecting AAAs formed by AngII-infusion in the analysis of section 2.
- The decision to focus only on ultrasound imaging for longitudinal, in vivo assessment of aneurysm progression makes sense as this is certainly the most common approach in the field. However, the authors have omitted discussion of volumetric ultrasound approaches in section 3.3 that alleviate many of the issues associated with probe position and operator dependence (10.1055/s-0041-1731404). Inclusion of 4DUS data further helps increase the ability of the user to assess vascular kinematics and strain mapping (doi: 10.1115/1.4043075).
- While the author’s choice of papers only after 2015 and only in ATVB could be justified in order to limit the scope of the literature analysis, older publications from other journals should almost certainly be included in section 3 when describing different imaging approaches. Possible addition modalities include:
- In vivo, pulse wave velocity ultrasound (doi: 10.1016/j.ultrasmedbio.2014.04.013)
- In vivo MRI - which the authors mention briefly but does not require contrast when time-of-flight angiography or black-blood spin echo sequences are used to image murine aneurysms, in contradiction to line 97 (doi: 10.1016/j.jacc.2011.09.017, 10.1161/CIRCIMAGING.108.787358).
- Ex vivo, phased contrast X-ray tomographic microscopy (doi: 10.1093/cvr/cvx128).
- Ex vivo, pressurized, panoramic digital image correlation.
- Ex vivo, pressurized, optical coherence tomography.
Minor Comments
- The term “imaging mode” is a little unclear. Does this refer to imaging modality such as MR, CT, or ultrasound? Or imaging mode – such as B-mode ultrasound? While this may be a question of semantics, I would suggest rephrasing to either “modality” or “technique”.
- Line 88: Suggest rephrasing “Aortic diameters were determined by measuring the longest distance between two points on the inner elastic laminae.” This is only true if the vessel is pressurized into a circle.
- Line 98: It would be beneficial to also include mention of the radiation dose associated with uCT. Indeed, high resolution scans with contrast will likely become more popular as the technology advances, but longitudinal studies where repeated very high resolutions scans with intravascular contrast is still not practically possible with current systems as the animals will eventually receive a lethal dose of radiation.
- Line 109: Suggest including reference to a previous meta-analysis where a review of the literature suggested both detection bias and publication bias in studies that have used the angiotensin II-infused dissecting AAA model (doi: doi.org/10.1093/cvr/cvv215).
- Figure 2 caption: recommend defining OCT in the caption.
- Line 178: The discussion of the tradeoff between ultrasound resolution and depth is somewhat misleading. High frequency ultrasound is ideal for mice not because of the shallow depth of the aorta, but because of the high resolution needed to image the vasculature.
- Line 181: Is 22-55 MHz refiring to transducer bandwidth? Or a range of center frequencies from a variety of transducers? Suggest clarifying.
- Line 186: A key paper to cite when discussing optimal ultrasound imaging techniques is from Trachet et al. in 2015 (doi: 10.1371/journal.pone.012900). The results suggest that “the short axis view is preferred over the long axis view to measure aortic diameters”.
- The authors may also want to consider pointing the reader to previous reviews on imaging of murine AAAs (doi: 10.2174/157016110793563898).
- The main point that it is optimal to combine both in vivo measurements with ex vivo quantification, even with simple imaging approaches, for validation of aortic geometry is great. Recommend highlighting this further in the final paragraph.
Reviewer 2 Report
The manuscript is a systematic review of Imaging Modes for Aortic Aneurysms and Dissections in Mice. The authors clearly state in Section 1, Introduction, the three approaches to be discussed: Ex vivo, in situ, and ultrasound.
Search strategy, eligibility criteria, and data extraction of available literature is clearly explained in Section 2, Preferred modes for aortic imaging in mice. Figure 1 clearly depicts the selection process of articles to be reviewed.
Section 3, Imaging approaches for aortic measurements in mice, discusses the three imaging modes with special emphasis on their strengths and weaknesses; while Section 4, Summary and Perspective, concludes that the optimal imaging approach would be a combination of in situ and ultrasound modalities.
The following suggestions are presented:
- Line 75 and Figure 1D: Change ‘29%’ to ‘28%’ in Figure 1D to match Line 75. The sum of the pie chart will effectively become 100%.
- Elaborate more on the ultrasound images presented (Figures 2C and 3)
- What systems were used?
- What types of transducer probes were used? (e.g. linear array, phased array, number of elements)
- What are the frequencies at which the images were acquired?
- Doppler echocardiography should be discussed more in depth in Section 3.3.3, Cardiac cycle. There are several articles dealing in detail with the correlation between cardiac cycle and Doppler echocardiograms.
- A comparison discussion could also be included regarding blood flow pattern differences between healthy and diseased mouse aortas.
Round 2
Reviewer 1 Report
Ito and colleagues have much improved their review manuscript with their additions and edits. They have addressed all comments and responded in detail.